# Effect of the Size of Titanium Particles Released from Dental Implants on Immunological Response

**DOI:** 10.3390/ijms23137333

**Published:** 2022-06-30

**Authors:** Juan Antonio Callejas, Javier Gil, Aritza Brizuela, Román A. Pérez, Begoña M. Bosch

**Affiliations:** 1Bioengineering Institute of Technology, Universitat Internacional de Catalunya, Josep Trueta s/n, Sant Cugat del Vallés, 08190 Barcelona, Spain; jacallejas@gmail.com (J.A.C.); rperezan@uic.es (R.A.P.); 2Facultad de Odontología, Universidad Europea Miguel de Cervantes, C. del Padre Julio Chevalier 2, 47012 Valladolid, Spain; aritzabrizuela@hotmail.com

**Keywords:** titanium, implantoplasty, dental implants, cytotoxicity, inflammatory reaction

## Abstract

The techniques used in oral implantology to remove bacterial biofilm from the surface of implants by machining the titanium surface (implantoplasty) or by placing rough dental implants through friction with the cortical bone generate a large release of particles. In this work, we performed a simulation of particle generation following clinical protocols. The particles were characterized for commercially pure titanium with particle sizes of 5, 10, 15, and 30 μm. The aim was to determine the effect of particle size and chemical composition of the implant on the immune response. For this purpose, their morphology and possible contamination were characterized by scanning electron microscopy and X-ray microanalysis. In addition, the granulometry, specific surface area, release of metal ions into the medium, and studies of cytocompatibility, gene expression, and cytokine release linked to the inflammatory process were studied. The release of ions for titanium particles showed levels below 800 ppb for all sizes. Smaller particle sizes showed less cytotoxicity, although particles of 15 μm presented higher levels of cytocompatibility. In addition, inflammatory markers (TNFα and Il-1β) were higher compared to larger titanium. Specifically, particles of 15 μm presented a lower proinflammatory and higher anti-inflammatory response as characterized by gene expression and cytokine release, compared to control or smaller particles. Therefore, in general, there is a greater tendency for smaller particles to produce greater toxicity and a greater proinflammatory response.

## 1. Introduction

Titanium dental implants have been successful in replacing teeth due to their osseointegration capability, good mechanical properties (especially in fatigue due to cyclic chewing loads), high corrosion resistance, and aesthetic properties in the mouth [1,2,3,4]. Every year, the number of patients using dental implants exceeds five million, and they are being increasingly used in other fields of medicine [5,6].

In a healthy human body weighing 70 kg, the amount of titanium does not exceed 15 mg [7]. These particles are generally located in hard and soft tissues; they are rather insoluble and are difficult to eliminate by the human body. In principle, at these concentrations, there is no significant immune response. However, if local concentrations are high, this can cause inflammatory processes around the tissues and cause variations in osteoblast and osteoclast concentrations, among other detriments [8,9,10].

Currently, new techniques in implantology are producing very important increases in titanium particles, for example, in the placement of dental implants at the bone level where the compression forces between the rough implant surface and the cortical bone cause the release of titanium particles into the environment [11,12]. Another case leading to a large release of titanium particles into the oral environment is the technique of implantoplasty for cases of peri-implantitis. This technique consists of the mechanization of the surface of the dental implant on which a bacterial biofilm is created, whereby the mechanical removal of the infected implant surface causes a large release of particles into the oral environment [13,14,15,16]. This technique avoids the replacement of the osseointegrated implant by eliminating the infection. Peri-implantitis affects more than 18% of placed dental implants, and this is why the technique of implantoplasty is being increasingly used by clinicians. Likewise, the placement of implants at a bone level represents approximately 37% of the total number of dental implants placed [17,18,19]. These facts suggest that titanium particle levels are increasing significantly in the human body; the immune response to these particles is not well known and has to be studied.

Furthermore, different authors have shown that titanium particles do not encapsulate in the tissue around the dental implant but migrate through the blood and can accumulate in different organs, causing systemic allergies and reactions [8,9,10].

In this work, particles of different sizes were isolated by simulating implantoplasty processes with more than 200 titanium dental implants, and they were classified into different mean sizes of 5 μm (Ti-5), 10 μm (Ti-10), 15 μm (Ti-15), and 30 μm (Ti-30). The reason for choosing these particle sizes is due to the fact that, in the literature, they are the most common ones found in the tissues after implantoplasty processes [20,21,22,23,24,25] and after the placement of bone-level dental implants caused by the friction of the collar on the cortical bone [26,27,28]. A limitation of this type of study is the difficulty in determining and characterizing particles smaller than 1 μm in size, although they can be observed in small quantities through field-emission electron microscopy observations and micro-CT characterization. However, the effect of these particles, which may not be detected by the immune system, should be studied.

## 2. Results

Our first characterization assay was to analyze the size of the microparticles, as shown in Figure 1. The average particle size for each classification can be observed in Table 1.

The specific surface areas are shown in Table 2. The results indicate that small samples had the highest specific surface area and were, therefore, the most reactive with the surrounding physiological environment. The difference between the specific surface areas of the Ti-5 μm and Ti-15 μm samples was not statistically significant. On the other hand, the Ti-30 μm sample had a significantly lower specific surface area than the other samples. That is, 30 µm particles had less reactivity per unit weight than the smaller particles. Consequently, we could compare the effect of the chemical composition on the immunological response.

As the particle size results show, no large differences in particle size were observed among the Ti-5 μm, Ti-10 μm and Ti-15 μm samples, but a significant difference was recorded for the 30 μm sample. What is apparent from the electron microscopy images is the regularity of the particle shapes and the homogeneity of the particle sizes (Figure 2). The energy-dispersive X-ray microanalysis involved 20 measurements for each of the samples, and only the peak typical of commercially pure titanium was visible. Therefore, we can say that the particles did not show any contaminants in their structure; in particular, residues of AISI 304 martensitic stainless steel from the drill or tungsten carbides were not observed.

The results of the ion release can be seen in Table 3. Typical ion release curves are suggested, with faster release kinetics in the first 1–3 days before the levels stabilized to a cumulative concentration between 575 and 800 ppb (μg/L) for the samples with a higher specific surface area (Ti-5), whereas samples with the smallest specific surface area (Ti-30) showed values from 485 ppb to 625 ppb after 21 days of immersion.

It can be seen from Table 3 that Ti-30 showed the significantly lowest ion release values in all cases.

A cytotoxicity test was performed at 24 and 48 h to assess cell survival. For this, THP-1 cells were cultured in medium containing the extracts at different concentrations (ISO 0.2 g/mL), as well as their 1:2, 1:10, 1:100, and 1:1000 dilutions. The cells were adherent to the substrate plate and presented the expected morphology, both before and after incubation with the extracts.

As can be seen in Figure 3, cells with ISO concentrations and their 1:2 dilutions with microparticles of sizes 10 μm and 30 μm showed a survival of less than 70% after 24 h culture, considered as cytotoxic. In contrast, at 48 h, the ISO concentration and its 1:2 dilutions were also considered cytotoxic in the 5 μm sample, whereas, in the culture of the 15 μm titanium microparticles (Ti-15), only the ISO concentration was considered cytotoxic.

Once the cytotoxicity of the extracts in THP-1 cell culture was assessed, the inflammatory response was tested at the concentration considered noncytotoxic. As can be seen in Figure 4, in the case of gene expression, the proinflammatory markers (CCR7, TNFα, and IL-1β) showed similar levels to the control (TCP), which were significantly lower when cultured with proinflammatory LPS medium, the positive control for inflammation (Figure 4A). Specifically, two proinflammatory markers (TNFα and IL-1β) presented significant lower values, mainly TNFα, at 48 h compared to the control sample, and this value decreased when the particle size increased. These values seemingly indicate that, as the size increased, the inflammation response decreased.

On the other hand, the anti-inflammatory markers (CD206, TGF-β, and IL-10) followed the same trend as the control sample (TCP) as they presented similar values, except for the values of the largest sample (Ti-30) which were higher than the control (Figure 4B). Moreover, when comparing the test samples at 48 h, our results show that smaller particles presented significantly lower values than bigger particles. Therefore, as seen for the proinflammatory genes, these results indicate that particles with bigger size had a greater anti-inflammatory response. In contrast, in the case of IL-1β interleukin, the values of the titanium (Ti-15) extracts were significantly lower at 48 h. These results indicate that the cells did not show an inflammatory response when cultured in medium conditioned with the samples.

Once the response was evaluated at the gene level, cytokine release analysis was performed. It can be seen in Figure 5 that the proinflammatory markers (TNFα and IL-1β) followed a similar trend to the control (TCP), and these levels were significantly lower when cultured with the proinflammatory LPS medium, the positive control of inflammation (Figure 5A). Specifically, at 48 h, the sample with the highest protein expression was the largest (Ti-30), exhibiting significantly higher values than almost all samples. On the other hand, the anti-inflammatory marker analyzed (IL-10) obtained similar results to the control sample (TCP), with the 10 μm and 30 μm samples (Ti-10 and Ti-30) presenting a lower value, being only significant in the case of Ti-10. These results indicate that the cells did not show an inflammatory response when cultured in the conditioned medium of the samples. Furthermore, these results are in agreement with those analyzed for cytotoxicity, where it can be observed that a larger size led to greater toxicity and a higher cytokine release of proinflammatory markers.

## 3. Discussion

Titanium is an excellent biomaterial for implants due to its excellent biocompatibility, osseointegration, corrosion resistance, and mechanical properties (static and cyclic) [29,30]. It has been widely used in the dental, maxillofacial, orthopedics, cardiovascular, and plastic surgery fields. Titanium can be damaged by machining, frictional forces, mechanical wear, and chemical corrosion in short- and long-term use, causing particle detachment. These particles vary in size, shape, and content in local and remote regions, destroying the bone homeostasis around the implant and further increasing the inflammatory response of surrounding living tissues, which can produce different diseases such as peri-implantitis [31,32] Methods to remove the particles or at least make them less harmful to the human body should be studied. Furthermore, the potential impact on other cells, tissues, and organs still needs to be explored.

The aim of this study was to evaluate the inflammatory response which occurs during implantoplasty. For this reason, commercially pure Ti particles (grade 4) were used as they resemble the detached particles that appear during this process and avoid the corrosion observed in the real procedure [33]. These particles were characterized, and then an in vitro assay was performed in order to evaluate the cytotoxicity.

Initially, Ti particles of 5, 10, 15, and 30 μm were characterized, showing expected average diameters (Table 1 and Figure 1) and irregular shapes (Figure 2). The specific surface area values indicated that smaller sizes were the most reactive with the surrounding physiological environment (Table 2). Moreover, the release levels were also analyzed, and it was determined that the ion release levels for Ti particles were always below 800 ppm (mg/L) (Table 3). This is in congruence with other studies that demonstrated how the higher specific surface area of particles is related to a higher release of molecules.

We then assessed the cytotoxicity of the extracts of Ti particles at 24 h and 48 h using the THP-1 cell line. This cell line has been widely used for evaluating immune response due to its genetic homogeneity, which facilitates the reproducibility of these experiments [34,35,36]. It is known that size is an important factor for the immunological response [37]. Our cytotoxicity studies showed that the lowest levels of cytotoxicity were recorded for the smallest particles, with 15 μm particles having the lowest cytotoxicity levels (Figure 3). It is well known that titanium ions are able to stimulate cell behavior by enhancing cell proliferation [38]. This allows us to consider that the difference in cell toxicity between the 15 µm and 30 µm particles is probably related to the decreased level of titanium ions in the 30 µm experimental group. The lowest levels of ion release were observed for the 30 µm particles. There was a threshold value above which the toxicity could be increased. The 5 and 10 µm particles were also nontoxic; however, the 15 µm particles were in the optimum range in terms of ion release [39,40,41,42,43,44,45].

These particles are generally proinflammatory and cytotoxicity, as they increase the inflammatory behavior of the tissues surrounding the dental implant, thus favoring peri-implantitis and producing bone destruction [46,47,48]. The lack of osteoblast activity in relation to the concentration of Ti debris was also confirmed by Happe et al. [43]. In order to evaluate how Ti debris promotes immune response, diverse interleukins and specific factors previously described for this analysis were studied [34].

Regarding the gene expression, results demonstrated that, as the size increased, the inflammation response decreased, especially at 48 h (Figure 4). Our results showed that particles of 15 μm presented a lower pro-inflammatory (TNFα and IL-1β) and higher anti-inflammatory response, compared to the control or smaller particles. This is in line with the cytotoxicity results, as the 15 μm particles were the least cytotoxic (Figure 3). LPS was used as a positive control, showing that our samples tended to have an anti-inflammatory response. However, the positive control LPS resulted in an increased expression of the marker IL-10. This is in congruence with other authors who previously described this effect, as some proinflammatory cells can express both TNFα and IL-10 markers [34,47].

In the case of cytokine release, the results were in agreement with those analyzed for gene expression cytotoxicity, where it could be observed that the 15 μm particles had the lowest cytotoxicity and proinflammatory properties.

In conclusion, our results show that Ti particles of 15 μm are biocompatible and present a lower immune response. Although this study presented a complete in vitro analysis of the effect of Ti particles during implantoplasty, following ISO 10993-5, further studies are needed in vitro to clearly understand the behavior of these particles. In this sense, a direct in vitro cell culture study should be performed in order to assess the effect of particle size, as well as a cell culture study with other cell types, mainly mesenchymal stem cells or osteoblasts, in order to predict the possible interaction and possible osseointegration of the materials and the effect of the particles in a real clinical scenario. Furthermore, future studies should include an in vivo scenario which could mimic the real procedure.

## 4. Materials and Methods

### 4.1. Dental Implants

Fifty commercially pure Titanium dental implants (Essential, Klockner, Escaldes Engordany, Andorra) (Figure 6) were mechanized by the same investigator (J.A.C.) according to the drilling protocol by Costa-Berenguer et al. [48], using a GENTLEsilence LUX 8000B turbine (KaVo Dental GmbH, Biberach an der Riß, Germany) and permanent water irrigation at 22 °C. The implant surface was sequentially mechanized with a fine-grained tungsten carbide burr (reference H379.314. 014 KOMET; GmbH & Co. KG, Lemgo, Germany), followed by a coarse-grained rubber polisher (order no. 9608.314.030 KOMET; GmbH & Co. KG, Lemgo, Germany) and finally by a fine-grained rubber polisher (order no. 9618.314.030 KOMET; GmbH & Co. KG, Lemgo, Germany). Due to the large amount of material required for all tests according to international standards, particles of the same size, of the same shape, and with very similar mechanical properties were purchased. Otherwise, the number of dental implants needed to obtain the particles would have exceeded 1000. To ensure that the particles were as close to reality as possible, we studied their morphology and morphometry using scanning electron microscopy. The nanohardness was used as the basis to most accurately simulate the real particles obtained in the experiment. The different particle sizes were supplied by Goodfellow (Goodfellow Cambrigde Ltd., Huntingdon, England).

The samples were lyophilized to remove water from the metal debris during the implantoplasty simulator process, allowing the physicochemical characterization and immunological study to be performed.

### 4.2. Specific Surface Area

The specific surface area, understood as the contact surface of the particle with the physiological medium, was determined using ASAP 2020 equipment (Micromeritics, Norcross, GA, USA) in vacuum conditions below 10 μmHg with nitrogen as the adsorbate. The particles were degassed at 100 °C. The specific surface area was determined by mathematical calculations according to the BET (Brunauer–Emmett–Teller) theory [49].

### 4.3. Granulometry

The size of the detached particles was determined using a Mastersizer 3000 (Malvern Panalytical, Malvern, UK). This unit uses the laser diffraction technique to measure particle size by measuring the intensity of scattered light as a laser beam passes through the sample of particles.

The test was performed in ethanol used as a liquid scattering medium. The equipment allows the analysis of particles between 9 nm and 3.45 mm. To avoid agglomeration of particles during the particle size test, two types of agitation were used: mechanical agitation using a 2500 rpm shaker and ultrasonic agitation at 50% sonication to ensure adequate measurement of all isolated particles.

### 4.4. Scanning Electron Microscopy

Particle morphology was investigated using a Jeol 6400 scanning electron microscope (JEOL, Tokyo, Japan) with a resolution of 15 nm and an acceleration voltage of 20 keV. Gold coating of the surfaces by sputtering was not necessary as the samples were sufficiently conductive. In addition, the microscope was coupled with an energy-dispersive X-ray microanalysis system (EDS Oxford, Oxford, UK).

### 4.5. Ion Release

Titanium ion release analyses were performed on five samples (*n* = 5) for each of the sizes according to ISO 10993-12-2009. For this purpose, a ratio of medium to solid particles of 1 mL per 0.2 g of particles used was used in accordance with the standard. In our study, 10 mL of medium was prepared for analysis, corresponding to 2 g of particles per test.

The liquid medium used for ion release was Hank’s saline solution (Sigma–Aldrich, Co., Life Science, St. Louis, MO, USA). Hank’s solution in contact with the particles was recovered and filtered through a filter with a pore size of 0.22 μm. For the analytical study, the solution was acidified with 2% nitric acid (HNO_3_ 69.99%, Suprapur, Merck, Darmstadt, Germany) to avoid precipitation of the metal ions prior to measurement of their concentration by inductively coupled plasma emission mass spectrometry (ICP-MS).

Extractions for analysis were performed at 1, 3, 7, 14, and 21 days following similar studies [50,51,52]. The samples were kept at 37 °C in an oven and shaken at 250 rpm, varying the inclination from 0° to 30° in order to avoid settling of the metal debris during the test and to ensure continuous exposure of all particles to the medium. The samples were analyzed by ICP-MS (Perkin Elmer Elan 6000, Perkin Elmer Inc., Waltham, MA, USA). This method permits quantitative multi-elemental analysis with an accuracy of 1 ppt (ng/L) for titanium.

### 4.6. Preparation of Samples for Cell Cultures

Each sample was independently sterilized with 96% ethanol prior to cell culture to determine cytocompatibility. Ethanol immersion of the samples lasted 30 min. Subsequently, the alcohol was removed by three cycles of centrifugation at 7200 rpm for 5 min each cycle. The samples were washed with Dulbecco’s phosphate-buffered saline (DPBS) (Sigma–Aldrich^®^, Sant Louis, MO, USA). Washing was repeated three times. In accordance with ISO 10993-5, cell culture medium was added after each centrifugation to maintain the concentration of 0.2 g of particles per mL of medium.

### 4.7. Cytotoxicity Assay

The cytotoxicity of the sample was evaluated by indirect exposure determination according to ISO 10993.

The THP-1 monocytic cell line was purchased from DSMZ (ACC 16). Cells were cultured and expanded in RPMI 1640 medium (Sigma) supplemented with 10% FBS (Sigma) and 1% penicillin–streptomycin (Fisher Scientific). They were cultured at a cell density of 3 × 10^5^ cells/mL.

For the cytotoxicity assay, cells were cultured on conditioned medium using titanium microparticles of different sizes (5, 10, 15, and 30 μm). The culture method was based on the indirect culture protocol of ISO 10993-5 “Biological evaluation of medical devices”(part 8.2). For this purpose, microparticles were cultured in THP-1 medium for 72 h in an incubator with 5% CO_2_ and 37 °C. Then, this conditioned medium was used at different concentrations to evaluate the cytotoxicity in the THP-1 cell culture. The concentration was selected according to ISO 10993-5, i.e., 0.2 g/mL, and we performed dilutions of this concentration (1:2, 1:10, 1:100, and 1:1000). Cytotoxicity was considered for levels beyond 70%, as indicated in ISO 10993-5.

The cytotoxicity assays were performed in triplicate (*n* = 3). The samples studied were test samples (Ti metal debris, one sample per size), positive control (cells seeded directly onto the plate), and negative control (medium without cells). Samples were handled aseptically throughout the assay.

### 4.8. Gene Expression Analysis

Gene expression was analyzed by quantitative real-time polymerase chain reaction (qPCR). Briefly, total RNA was isolated using a NucleoSpin RNA kit (Macherey–Nagel, Düren, Germany), which included DNAse treatment, following the manufacturer’s instructions. One microgram of RNA with a ratio of intensities at the wavelengths of 260/280 nm between 1.8 and 2 was then reversed-transcribed into cDNA using the Transcriptor First-Strand cDNA Synthesis Kit (Roche, Basel, Switzerland) according to the manufacturer’s recommendations. Specific primers for inflammatory response and FastStart Universal SYBR Green Master (Roche, Basel, Switzerland) were used to amplify the desired cDNA. As shown in Table 4, these primers were proinflammatory markers (CCR7, TNFα, and IL-1β genes) and anti-inflammatory markers (CD206, TGF-β, and IL-10 genes). Gene expression was normalized to the constitutive β-actin gene (housekeeping gene). Finally, the amplifications were performed in a CFX96 Real-Time PCR Detection System (Bio-Rad, Hercules, CA, USA).

### 4.9. Cytokine Release Analysis

Cell culture supernatants were collected at 24 and 48 h in order to quantify the release of cytokines by THP-1 cells. Two proinflammatory (TNFα and IL-1β) and one anti-inflammatory (IL-10) cytokines were analyzed. LPS was used as a positive control of inflammation, while medium without particles was used as a negative control. Quantification was performed using commercially available ELISA kits (Thermofisher scientific, Waltham, MA, USA) following the manufacturer’s recommendations.

### 4.10. Statistical Analysis

Data were recorded using a Microsoft Excel spreadsheet (Microsoft^®^, Redmond, Washington, DC, USA) and subsequently processed with the Stata 14 package (StataCorp^®^, College Station, San Antonio, TX, USA). Means and standard deviations were calculated, except for the granulometry test, where the mode and percentiles were used.

## 5. Conclusions

It was determined that particles with an average diameter of 5 μm had a higher specific surface area, which decreased with the increase in diameter. The shape of the particles was irregular and the release of titanium ions into the physiological medium at different times was higher for the particles with a smaller diameter, reaching 800 ppb after 21 days of immersion. The smallest ion release values corresponded to the 30 µm size. Particles of 15 μm presented higher levels of cytocompatibility. In addition, inflammatory markers (TNFα and Il-1β) were higher compared to the larger titanium particles. Specifically, particles of 15 μm presented a lower proinflammatory and higher anti-inflammatory response as analyzed by gene expression and cytokine release, compared to control or smaller particles. Therefore, in general, there was a greater tendency for smaller particles to produce greater toxicity and a greater proinflammatory response.

## Figures and Tables

**Figure 1 ijms-23-07333-f001:**
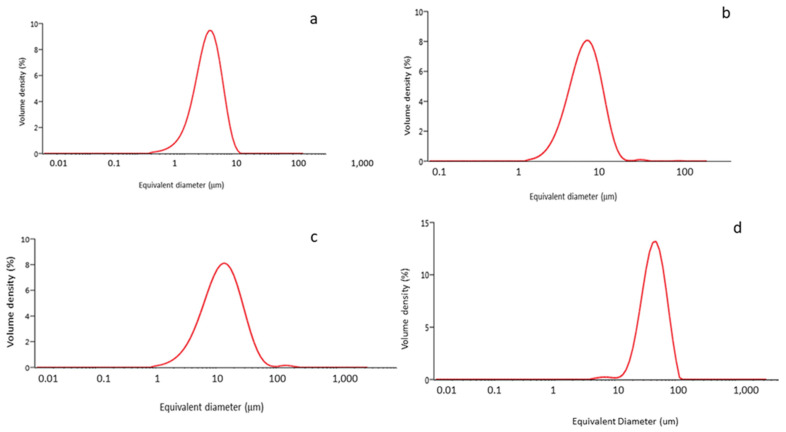
Granulometry curves for each class of particles: (**a**) 5 μm; (**b**)10 μm; (**c**) 15 μm; (**d**) 30 μm.

**Figure 2 ijms-23-07333-f002:**
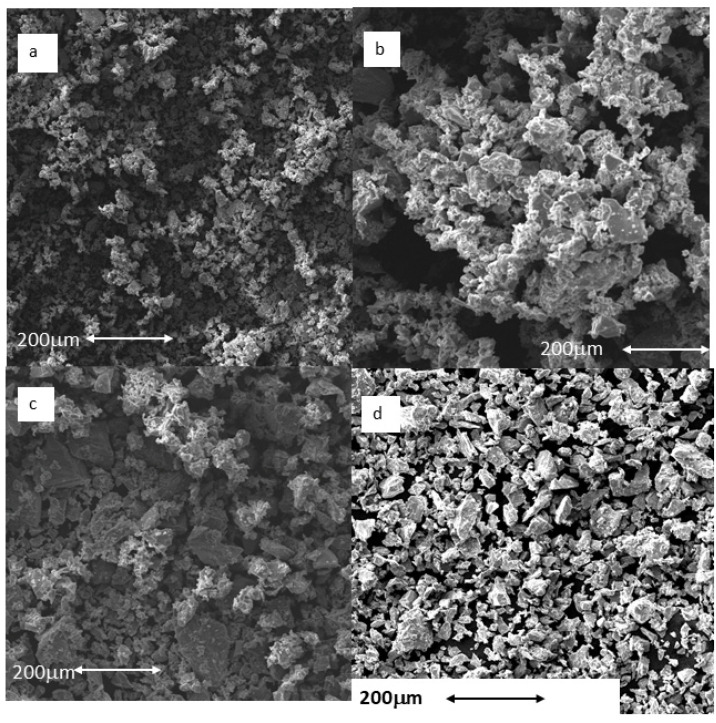
Particles observed by scanning electron microscopy: (**a**) Ti-5 μm; (**b**) Ti-10 μm; (**c**) Ti-15 μm; (**d**) Ti-30 μm. Scale bar = 200 μm.

**Figure 3 ijms-23-07333-f003:**
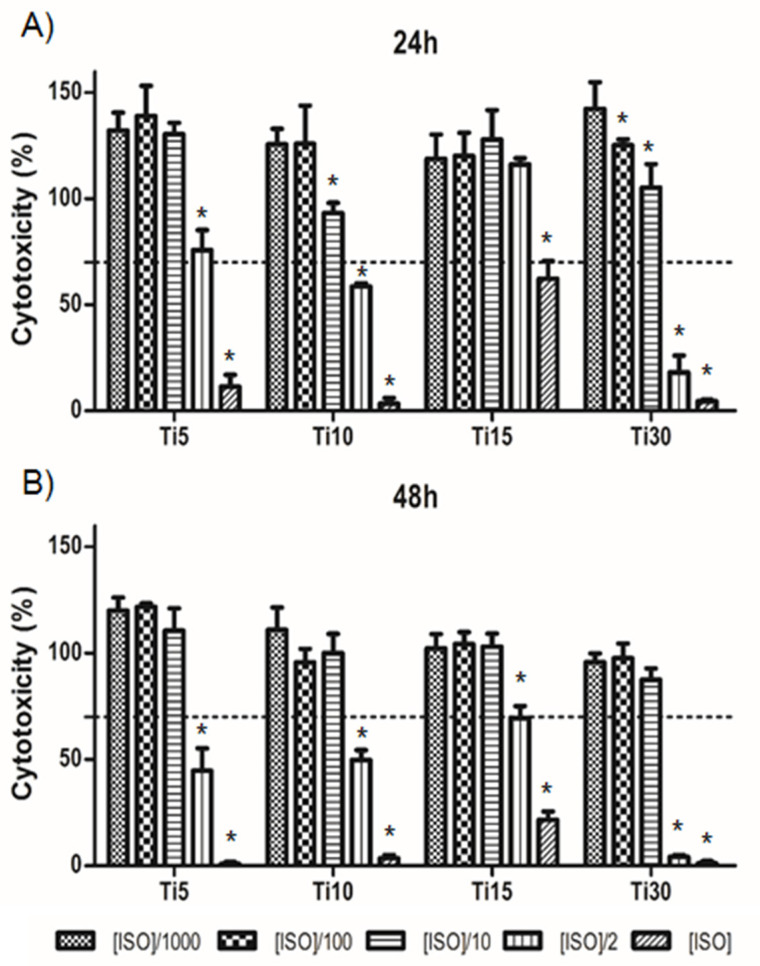
Cytotoxicity levels for THP-1. Cells cultured with titanium microparticle extracts at 5 μm, 10 μm, 15 μm, and 30 μm, performed at 24 h (**A**) and 48 h (**B**). Cells were cultured at ISO concentrations of 0.2 g/mL and its dilutions. * *p* < 0.05 vs. ISO/1000.

**Figure 4 ijms-23-07333-f004:**
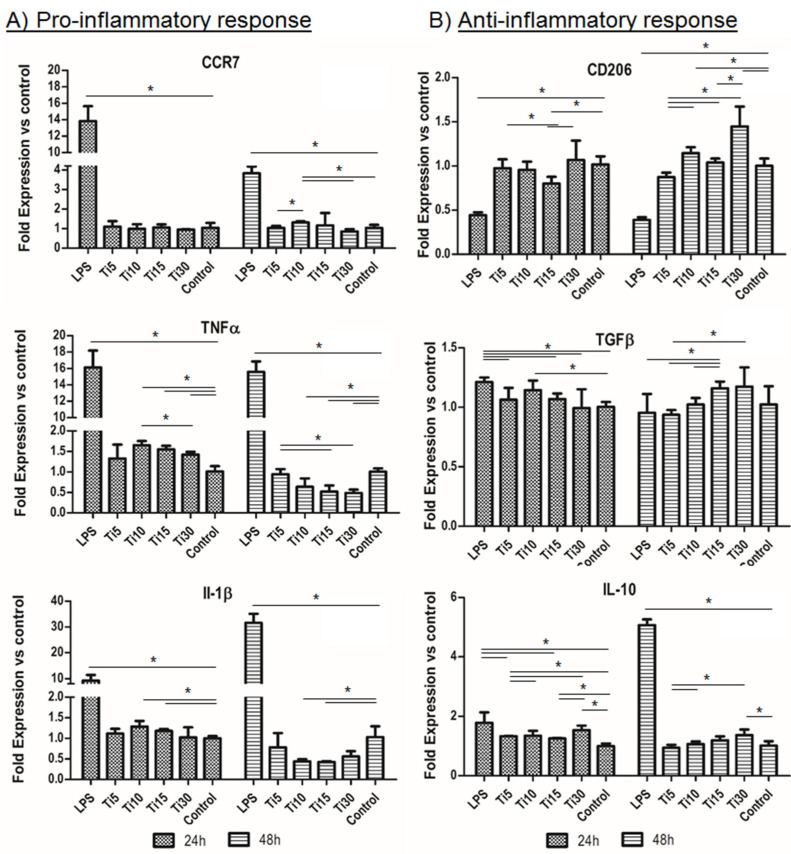
Immune response RNA analysis. RNA analysis of the inflammatory response in THP-1 cells from proinflammatory markers CCR7, TNFα, and IL-1β (**A**) and anti-inflammatory markers CD206, TGF-β, and IL-10 (**B**). Cultured cells with extracts of titanium microparticles at 5 μm, 10 μm, 15 μm, and 30 μm were analyzed at 24 h and 48 h. LPS was used as a positive control of inflammation, while medium without particles was used as a negative control. * *p* < 0.05.

**Figure 5 ijms-23-07333-f005:**
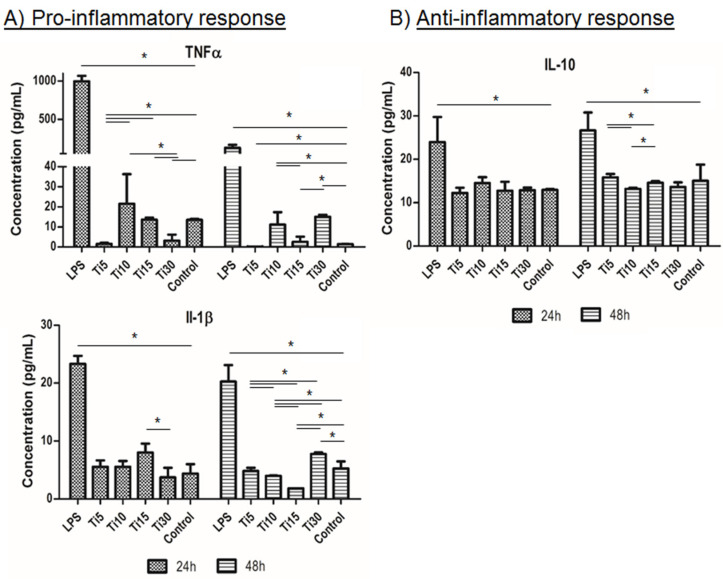
Immune response cytokine analysis. Cytokine release analysis of the inflammatory response in THP-1 cells from proinflammatory markers TNFα and IL-1β (**A**) and anti-inflammatory marker IL-10 (**B**). Cultured cells with extracts of titanium microparticles at 5 μm, 10 μm, 15 μm, and 30 μm were analyzed at 24 h and 48 h. LPS was used as a positive control of inflammation, while medium without particles was used as a negative control. * *p* < 0.05.

**Figure 6 ijms-23-07333-f006:**
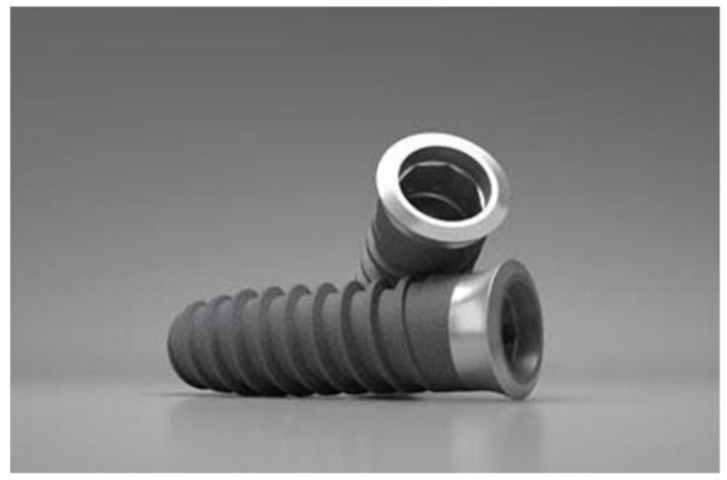
Dental implant studied. This implant was manufactured from grade 4 titanium.

**Table 1 ijms-23-07333-t001:** Average equivalent diameters for each class of particles studied.

Samples	Average Equivalent Diameter (μm)
Ti-5 μm	5.9
Ti-10 μm	9.7
Ti-15 μm	14.7
Ti-30 μm	30.3

**Table 2 ijms-23-07333-t002:** Specific surfaces for the different size of samples.

Samples	Specific Surface (m^2^/g)
Ti-5 μm	0.5124 ± 0.0234
Ti-10 μm	0.4888 ± 0.0342
Ti-15 μm	0.4702 ± 0.0119
Ti-30 μm	0.2001 ± 0.0589

**Table 3 ijms-23-07333-t003:** Titanium ion release in ppb for the different samples studied after 1, 3, 7, 14, and 21 days of immersion.

Time/Samples	Ti-5 μm	Ti-10 μm	Ti-15 μm	Ti-30 μm
1 day	575 ± 12 *°	525 ± 10 *°	508 ± 10 **°	485 ± 15 ***°
3 days	715 ± 19 *°°	701 ±12 *°°	650 ± 12 **°°	505 ± 12 ***°
7 days	725 ± 21 *°°	718 ± 23 *°°	700 ± 23 *°°°	575 ± 13 **°°
14 days	796 ± 10 *°°°	752 ± 17 **°°	732 ± 11 **°°°	612 ± 19 ***°°°
21 days	800 ± 18 *°°°	777 ± 15 **°°°	755 ± 15 **°°°°	625 ± 10 ***°°°

* Statistically significant differences in titanium ion release for each particle size and for the same immersion time (*p* < 0.01). ° Statistically significant differences in titanium ion release for the same particle size with different immersion times (*p*-value < 0.01). Different number of * means the statistical differences between those results. The same meaning is with ° symbol.

**Table 4 ijms-23-07333-t004:** Sequences of primers used for quantitative real time polymerase chain reaction.

Inflammatory Character	Gene	Forward (Sequence 5′–3′)	Reverse (Sequence 5′–3′)
Proinflammatory	TNFα	TTCCAGACTTCCTTGAGACACG	AAACATGTCTGAGCCAAGGC
	IL-1β	GACACATGGGATAACGAGGC	ACGCAGGACAGGTACAGATT
	CCR7	GGCTGGTCGTGTTGACCTAT	ACGTAGCGGTCAATGCTGAT
Anti-inflammatory	IL-10	AAGCCTGACCACGCTTTCTA	ATGAAGTGGTTGGGGAATGA
	TGF-β	TTGATGTCACCGGAGTTGTG	TGATGTCCACTTGCAGTGTG
	CD206	CCTGGAAAAAGCTGTGTGTCAC	AGTGGTGTTGCCCTTTTTGC
Housekeeping gene	Β-actin	AGAGCTACGAGCTGCCTGAC	AGCACTGTGTTGGCGTACAG

## Data Availability

The authors can provide details of the research upon request by letter.

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
