# Peer review of "Effect of the Size of Titanium Particles Released from Dental Implants on Immunological Response"

_ijms, 2022, doi:10.3390/ijms23137333_

Round 1

Reviewer 1 Report

Review comments of ijms-1764903 manuscript

The present manuscript was entitled “Effect of the size titanium particles released from dental implants on immunological response”. Authors have demonstrated with the higher level of cytocompatibility and subsequently, lower pro-inflammatory and higher anti-inflammatory response was observed in the immune responsive analysis employed with 15 um size Ti nanoparticles with the potential experimental demonstrations. Eventually, I would recommend this manuscript for publication after fulfilling the following major comments.

1.      Figure 1, the particle sizes of the 1a, 1b and 1c SEM images were shown similar particle sizes as compared with 1d image. So, the SEM data was not supported by the title of the current manuscript.

2.      In table 1, The Ti particle size distribution representation is confusing to the readers and in order to avoid that, it would be better to present the data along with the particle size distribution plot to the respective different sizes of Ti particles.

Author Response

Dear Reviewer,

Thanks for taking the time to review our manuscript and suggest to us to improve our work by providing a lot more detail. We have done so, and we are now submitting a manuscript that not only addresses the points that you specifically raised but also many others that we have considered in order to deliver what we think is a much improved version of our work. This version includes more paragraphs, English grammar revisions in all main sections and new references. Thanks a lot. We are looking forward to your comments.

Sincerely,

Javier Gil

REVIEWER 1

The present manuscript was entitled “Effect of the size titanium particles released from dental implants on immunological response”. Authors have demonstrated with the higher level of cytocompatibility and subsequently, lower pro-inflammatory and higher anti-inflammatory response was observed in the immune responsive analysis employed with 15 um size Ti nanoparticles with the potential experimental demonstrations. Eventually, I would recommend this manuscript for publication after fulfilling the following major comments. 

  1. Figure 1, the particle sizes of the 1a, 1b and 1c SEM images were shown similar particle sizes as compared with 1d image. So, the SEM data was not supported by the title of the current manuscript.

We have obtained new images of the particles by scanning electron microscopy (SEM) and a new figure has been introduced. One aspect to take into account is that the smaller particles agglomerate due to surface electrostatics and the larger ones do not have such an important effect. The authors believe that in this new figure the morphologies of the particles and an estimation of the size difference can be observed. Many thanks to the reviewer for this indication which has allowed us to improve the figure.

  1. In table 1, The Ti particle size distribution representation is confusing to the readers and in order to avoid that, it would be better to present the data along with the particle size distribution plot to the respective different sizes of Ti particles.

According to the reviewer, a new figure with the granulometry curves for each size family has been added. In this way, the dispersion of the particle size values can be better appreciated and the results can be better interpreted.

Reviewer 2 Report

The manuscript assesses the biological effect of titanium particles on cell response and immunological response. The topic and results are interesting for readers and clinical application. Some revision is suggested.

1. Please explain the ppb, ppt, or ppm unit for ions release.

2. Table 3. Is any statistical analysis for the ions release result among the experimental groups.

3. How to define and decide to use 5, 10, 15, and 30 micrometers as the experimental groups rather than 3 and 50 micrometers?

4. 15-micrometer titanium particles seem to be the best biocompatibility rather than 30-micrometer titanium particles. How to explain the findings?

5.  Figure 3, the expression of TGF-beta shows different tendencies between 24 and 48 hours time point. How to explain the findings? 

6. Inconsistnce is demonstrated among the results of IL- 1beta, IL-10, TNF-alpha, and IL-1beta. Please explain and clarify.

7. Please add the reference (J Mech Behav Biomed Mater. 2020 Oct;110:103899) related to the biocompatibility of Titanium in discussion line 157.

8.  Please extend the limitation of this study in discussion line 210.

Author Response

Dear Reviewer,

Thanks for taking the time to review our manuscript and suggest to us to improve our work by providing a lot more detail. We have done so, and we are now submitting a manuscript that not only addresses the points that you specifically raised but also many others that we have considered in order to deliver what we think is a much improved version of our work. This version includes more paragraphs, English grammar revisions in all main sections and new references. Thanks a lot. We are looking forward to your comments.

Sincerely,

Javier Gil

REVIEWER 2

The manuscript assesses the biological effect of titanium particles on cell response and immunological response. The topic and results are interesting for readers and clinical application. Some revision is suggested.

  1. Please explain the ppb, ppt, or ppm unit for ions release.

Thank you for the appreciation. The authors have defined in the text:

  • ppm: mg/L
  • ppb: um/L
  • ppt: ng/L
  1. Table 3. Is any statistical analysis for the ions release result among the experimental groups.

We have added the results in Table 3 on the statistically significant differences of titanium ion release as a function of particle size with constant immersion time and on the other hand with the same particle size and different immersion times.

“Statistical differences are shown with * for statistically significant differences of titanium ion release in relation to each particle size and for the same immersion time with a p<0.01. It can be seen that Ti 30 has significant differences in all cases showing the lowest ion re-lease values. The symbol o shows the statistically significant differences with a p-value <0.01 for the same particle sizes with different immersion times. In all cases, Ti30 due to the maximum specific surface of the particles presents the lowest titanium ion release.” 

  1. How to define and decide to use 5, 10, 15, and 30 micrometers as the experimental groups rather than 3 and 50 micrometers?

Thank you very much for your suggestion, the authors have added a paragraph explaining the reason for the choice of these particle sizes and several bibliographic citations have been included to expand on this aspect.

“The reason for choosing these particle sizes is due to the fact that in the literature they are the most common ones found in the tissues after implantoplasty processes [20-25] and due to the placement of bone level dental implants caused by the friction of the collar on the cortical bone [26-28]. A limitation of this type of study is the difficulty in determining and characterizing particles smaller than 1 micrometer in size, although in Field Emission Electron Microscopy observations and by micro-CT characterization they are observed in small quantities. However, the effect of these particles, which may not be detected by the immune system, should be studied.”

  1. 15-micrometer titanium particles seem to be the best biocompatibility rather than 30-micrometer titanium particles. How to explain the findings?

It is indeed an interesting point. We have a similar thinking regarding this issue. From our understating, we visualize both 15 and 30 microns to have the same size and hence should behave in a similar manner. Similar finding was expected in the 30-micron size particles. We understand the 15-micron result does not follow the trend that the 5 and 10 microns particles showed and that was continued by the 30 micron particles. We believe that this anomaly in the results, which was repeated to verify the result, is probably related with the amounts of ions present in the culture medium. Ti and TiO2 which is was is being formed in the culture medium are known to eventually enhance cell proliferation. At 24 h, the 5, 10 and 15 microns particles show nearly nontoxic results for ISO/2 whereas 30 microns particles show toxicity at this concentration. The trend is more or less similar at 48, once again having lower toxicity levels for the 30 micron particles. This is probably implying that the release of ions is necessary for the cell proliferation, which is ameliorated with their presence, although the levels of therapeutic level and toxicity levels, have a small window.

We have added a sentence and a reference to support the data.

“It is well known that titanium ions are able to stimulate cell behavior, enhancing cell proliferation in the presence of this ion (REF). This allows us to consider that the difference in cell toxicity between the 15 micron and 30 microns is probably related with a decreased level of titanium ions in the 30-micron experimental group. As can be seen from the ion release, these have the lowest levels for the 30 micron particles. In a similar way, there is also a threshold value above which, the toxicity can be increased The 5 and 10 micron sized groups are also on the non-toxic levels, as the results show low levels of toxicity, although the 15-micron size particles seem to be in the optimum range in terms of ion release.”

  1. Figure 3, the expression of TGF-beta shows different tendencies between 24 and 48 hours time point. How to explain the findings? 

We agree with the reviewers that 24 h and 48 h show a different trend. We have analyzed and revised these results over and over, even with experts in the field in order to understand. Our conclusion is that despite there are differences in the trend, we understand that the variances, even being significant when comparing different groups, are in the range of slight differences and hence with did not consider them significantly enough to strive conclusions from these results. The levels for all groups are in the range of 1 and the differences between the same groups at different time points, are not significant in all cases.

  1. Inconsistnce is demonstrated among the results of IL- 1beta, IL-10, TNF-alpha, and IL-1beta. Please explain and clarify.

We thank the reviewer once again for raising this point. We understand that there can seem to be inconsistencies. We have carefully analyzed the results once again. Figure 3 corresponds to gene expression whereas Figure 4 corresponds to protein analysis of the secreted proteins. Taking this into account and considering that experiments were performed at 24 and 48 hours, it is plausible to consider that the gene expression is probably more reliable. The reason is that first a change in the gene expression is needed in order to induce a change in the protein expression level, despite the processes can be fast, we consider that it is possible that at 48 hours, the protein levels have not raised or have not been affected sufficiently in order to observe the change. There seem to be a trend for the protein expression, showing that at 24 hours the expression of pro-inflammatory factors is higher for the smaller particles, probably related with a burst of ions that can induce the pro-inflammatory activity. Once the initial burst is decreased, the pro-inflammatory factors decrease whereas the 30-micron group show increased levels. On the other hand, the anti-inflammatory does not seem to show a clear trend. Hence, protein expression, which we know is later than gene expression shows higher inflammatory response for the smaller particles at short times, followed by higher pro-inflammatory effect of the bigger particles at later time points.

The trend is more or less similar in the pro-inflammatory molecules analyzed by gene analysis. It does seem that at 24 hours the trend seems to be similar, although not so markedly distinguished. At longer time points, 48 hours, all values seem to decrease, showing a similar effect as the one shown in the protein expression levels. In general, both at 24 and 48 hours, the expression for gene analysis is nearly close to 1 and hence no expansive differences are expected, which is more or less followed in the protein expression levels. Regarding the anti-inflammatory molecules, the trend is similar, having similar values in all case for the gene expression, which is a trend similar to the one in the protein expression.

  1. Please add the reference (J Mech Behav Biomed Mater. 2020 Oct;110:103899)related to the biocompatibility of Titanium in discussion line 157.

Thank you for the suggestion. We have added the reference in the discussion part.

  1. Please extend the limitation of this study in discussion line 210.

The section has been extended as follows:

“further studied are needed in vitro to clearly understand the behavior of these particles. In this sense, a direct in vitro cell culture study should be performed in order to assess the effect of particle size, as well as a cell culture study with other cell types, mainly mesenchymal stem cells or osteoblasts, in order to predict the possible interaction and possible osseointegration of the materials and the effect of the particles in real clinical scenario. Furthermore, future studies should include an in vivo scenario which could mimic the real procedure.”

Round 2

Reviewer 1 Report

Authors have been improvised the revised manuscript accordingly,  So the current revised manuscript, I would recommend for publication without further revision.

Reviewer 2 Report

Accept the revised manuscript.